# Individualized diagnosis of rheumatoid arthritis: A rank-based qualitative T cell-related signature

**Hang Su**[1][◐], **Xingyi Li**[1][◐], **Yawei Li**[2]*

1 School of clinical medicine, Guizhou Medical University, Guiyang, Guizhou, China, 2 School of Biology and Engineering, Guizhou Medical University, Guiyang, Guizhou, China

◐ These authors contributed equally to this work

* liyawei@gmc.edu.cn

## Abstract

Rheumatoid arthritis (RA) is a systemic autoimmune disease with persistent synovitis and joint destruction, leading to a huge economic and physical burden on patients. The detection of RA is important for the individual's guiding therapeutic. However, current signatures lacked enough effects for the diagnosis of RA. Here, a pariwise signature, including genes *ICAM2* and *OSTF1*, was derived based on a rank-based method, which was called *ICAM2-OSTF1* signature (IOS). The sensitivity and specificity of IOS in the training dataset were 87.39% and 86.79%, respectively. The accuracy of IOS was 91.07% in the validation dataset that contained a total of 280 samples from two independent datasets. Besides, when using eight methods, such as ssGSEA, xCell and TIMER, to quantitate the immune infiltration characteristics in RA. We found that RA presented elevated pro-inflammation immune infiltration and immune score. In addition, transcriptome analysis demonstrated that the consistent transcriptional differences between RA and healthy control were significantly enriched in some pathways typically related to the immune microenvironments, such as T cell activation. Finally, network analysis demonstrated that *ICAM2*, *CXCL16*, *CKLF* and *SLPI* may be related to the occurrence of RA. In brief, IOS can individually distinguish RA from healthy controls measured by different laboratories, and be an auxiliary test for diagnosing RA.

## Introduction

Rheumatoid arthritis (RA) is a systemic autoimmune disease, with an incidence of 0.5 to 1% in the world [1]. With the development of the disease, the inflammation caused by RA leads to irreversible joint damage, disability and even a shortened lifespan [2,3]. Fortunately, evidence suggested that early detection of RA can effectively reduce the degree of joint destruction, and associated problems in about 90% of patients [4]. Currently, serum biomarkers used to detect RA in clinical settings lack enough accuracy, including serum rheumatoid factor and anti-cyclic citrullinated

**Data availability statement:** The datasets analyzed during the present study are available from the Gene Expression Omnibus (https://www.ncbi.nlm.nih.gov/geo/query/acc.cgi?acc=GSE17755;https://www.ncbi.nlm.nih.gov/geo/query/acc.cgi?acc=GSE97475;https://www.ncbi.nlm.nih.gov/geo/query/acc.cgi?acc=GSE97810).

**Funding:** This research was supported by a grant from the Basic Research Program of Guizhou Science and Technology Department (ZK[2024]156) and the High-level Talents Startup Fund of Guizhou Medical University (J[2022]042. The funder (Yawei Li) conceived the idea of this study and helped in interpreting the results and writing.

**Competing interests:** The authors have declared that no competing interests exist.

peptide antibody [5]. Therefore, it is urgent to develop an accurate molecular signature for the detection of RA, thereby improving the patient's prognosis.

Many transcriptional signatures for diagnosing RA have been provided [6–9]. However, a critical limitation of such quantitative signatures is that their applications are commonly based on the quantitative measurements of the signature genes, which are impractical for clinical applications due to the considerable variance in the expression levels of the same gene across samples [10]. In contrast, signatures developed based on within-sample relative expression orderings (REOs) of gene pairs are resistant to batch or normalization effects by different technical sources and can be applied to individual samples [11–13].

In addition, T cells play critical roles in RA pathogenesis by producing and releasing various inflammatory mediators, including chemokines and cytokines [14]. And T cells are also related to inflammatory responses and interferon responses [15]. For example, regulatory T cells are involved in maintaining immune balance and tolerance to self-antigens, whose dysfunction contributes to RA pathogenesis [16,17]. Type 1 T helper cells also are highly activated in RA patients, which have the ability to secrete pro-inflammatory factors such as interferon-gamma [18]. Moreover, as a part of the innate immune system, natural killer T cells significantly decrease in the peripheral blood of RA patients when compared with healthy controls [19,20]. In total, T cells are implicated in RA occurrence and development by triggering long-term pathological chronic inflammation. Therefore, T cell-related genes may have the potential ability for diagnosing RA.

In the present study, we used the expression profiles of T cell-related genes, including regulatory T cells, T helper cells and natural killer T cells, to extract a classification biomarker. Quantitative measurements of the same gene are highly varied across different samples, even from the same type. Thus, we applied an REO-based method to discover a qualitative signature for the classification of RA by transforming the quantitative expression of a single gene into a binary relation of the within-sample REOs of gene pairs. The performance of the signature was validated in two independent datasets.

## Materials and methods

### Data collection and preprocessing

Three datasets of RA and normal peripheral blood samples profiled by different microarray platforms were downloaded from Gene Expression Omnibus (GEO, http://www.ncbi.nlm.nih.gov/geo/). As summarized in Table 1, GSE17755 dataset measured

**Table 1. The datasets analyzed in this study.**

| Dataset | normal | RA | sample | Platform |
|---|---|---|---|---|
| GSE17755 | 53 | 112 | PBMC | Hitachisoft AceGene Human Oligo Chip 30K 1 Chip Version |
| GSE97475 | 33 | – | PBMC | Illumina HumanHT-12 V4.0 expression beadchip |
| GSE97810 | – | 247 | PBMC | Illumina HumanHT-12 V4.0 expression beadchip |

Abbreviations: RA, Rheumatoid arthritis; PBMC, peripheral blood mononuclear cell

by the Hitachisoft AceGene Human Oligo Chip 30K 1 Chip Version (GPL1291) was used as the training dataset, which includes 53 normal and 112 RA peripheral blood samples. The 33 normal samples from GSE97475 dataset, together with the 247 RA samples from GSE97810 dataset, were used as the validation dataset.

The identical preprocessing was executed for these datasets, with the gene expression profiles (series.txt) utilized directly following the probe re-annotation process, devoid of any normalization. The probe set IDs were correlated with Entrez gene IDs using the relevant platform annotation file for all gene expression profile data. If numerous probe sets corresponded to the same gene, the expression value was determined by calculating its average value. Furthermore, probes that mapped multiple Entrez gene IDs or failed to map any Entrez gene ID were eliminated.

## Development of the REO-based signature

A total of 233 T-related genes were downloaded from TISIDB database [21] (http://cis.hku.hk/TISIDB/), including 36 T follicular helper cell-related genes, 77 type 1 T helper cell-related genes, 27 type 17 T helper cell-related genes, 29 type 2 T helper cell-related genes, 20 regulatory T cell-related genes and 64 natural killer T cell-related genes.

Using T cell-related gene expression profiles, a binary matrix was produced by pairwise comparisons of all genes, with $E_i > E_j$ or $E_i < E_j$ on an individual sample. $E_i$ and $E_j$ were used to represent the expression values of gene $i$ and gene $j$, respectively. Then, highly stable gene pairs were defined as those whose REO relation was consistent at ≥ 85% of samples. We detected highly stable gene pairs in RA and normal samples from the training dataset, respectively. Finally, the overlap of these two pairwise sets was considered as the REO-based signature.

## Identification of immune cells infiltration in RA

The immune infiltration of 28 immune cell types was also evaluated by ssGSEA method, whose signature genes were derived from previous report [22].Transcriptome-based algorithms XCELL [23], TIMER [24], QUANTISEQ [25], MCP-COUNTER [26], EPIC [27] and CIBERSORT [28] were used to estimate the abundance of immune cells infiltration in RA, including 28 immune cell types. Immune score was estimated by Estimate method [29]. Spearman's rank correlation was used for evaluating the relationship between signature and T cell infiltration.

## Differential analysis

The Wilcoxon rank-sum test was used to detect differentially expressed genes (DEGs) between normal and RA samples from training and testing datasets. The BenjaminiHochberg (BH) procedure was used to calculate false discovery rate (FDR). The significance was assessed by FDR value of 0.01. Further, the overlaps with consistent dysregulation between these two gene lists were considered consistent DEGs. Inflammatory-related gene lists were obtained from the Gene-Cards database (https://www.genecards.org/) utilizing four keywords: "Interferon response," "chemokine," "Cytokine," and "Inflammatory response," under the categories "Protein Coding" and "Proteins" (S1 Table).

## Pathway enrichment analysis

Functional enrichment analysis of DEGs and genes in network modules was performed using the R package 'clusterPro-filer' [30]. Subsequently, histogram and chordal maps of the Gene Ontology (GO) and Kyoto Encyclopedia of Genes and Genomes (KEGG) were drawn.

## Analysis of immunity features

The one-sided Wilcoxon rank-sum test was employed to compare the aforementioned immunity features between normal and rheumatoid arthritis samples. Spearman rank correlation was used to assess the relationships between the expression levels of two signature genes and the immune infiltration of 28 immune cells.

## Network analysis

Based on the expression and profiles of testing dataset, Spearman rank correlation, with $|r| > 0.6$ and FDR $< 0.01$, was applied to identify DEGs that were co-expressed with signature genes and inflammation-related genes.

## Statistical analysis

All statistical analyses in this study were performed using R software versions 4.0.2 (http://www.r-project.org/).

# Results

## Development of REO-based diagnostic signature

The discovery workflow is shown in Fig 1. Here, we identified 2,775,389 highly stable gene pairs in 112 RA samples and 808,065 gene pairs in 53 healthy samples (see Materials and methods), wherein REOs of 25,185 candidate gene pairs were stable in healthy samples but reversed in RA samples. Then, a total of 233 T cell-related genes (TRGs) from the TISIDB database were obtained and 25,185 T cell-related gene pairs (TRGPs) were established from these genes by the discretization operation. Only one gene pair, *ICAM2-OSTF1*, overlapped between TRGPs and reversed gene pairs and was considered the diagnostic signature, called *ICAM2-OSTF1* signature (IOS). According to a rule of "winner takes all", we then directly used IOS to classify samples in the training dataset. A sample was classified into the RA group if the REO of IOS voted for RA ($E_i < E_j$); else, the sample was classified into the healthy group ($E_i > E_j$).

## Performance of IOS in the training and independent datasets

For peripheral blood samples in the training dataset (GSE17755) with 112 RA samples and 53 healthy samples, the AUC of IOS reached 87.10% (95% CI, 81.50–92.60%, Fig 2A). 97 out of 112 RA samples and 46 out of 53 healthy samples were accurately categorized. The accuracy of IOS was 87.20% (Fig 2B). The performance of the signature was evaluated using the union of the datasets GSE97475 and GSE97810. As shown in Fig 2C-D, the signature resulted in a 92.3% AUC (95% CI, 87.80–96.80%), and a prediction of RA samples with a 90.69% sensitivity, 93.94% specificity, and 91.07% accuracy. Further, as shown in Fig 2E, the fold change of the expression levels between *ICAM2* and *OSTF1* took values ranging from 0.96 to 1.20 with the median of 1.07 in the normal samples, while in the RA samples the fold change took values ranging from 0.85 to 1.06 with the median of 0.96 (Fig 2F). Similar results were observed in training dataset (S1 Fig), indicating that the fold changes of IOS are quite different across different samples for each of the two subtypes but the REO patterns of IOS are stable.

## RA sample presenting elevated immune infiltration

The occurrence and development of RA are associated with immunological microenvironments [31]. Utilizing the ssGSEA approach, the abundance of 28 infiltrating immune cell types was illustrated using 280 samples from the testing datasets to investigate the correlation between RA and microenvironments. The results indicated that RA exhibited a greater abundance of immune cell infiltration, including activated dendritic cells, activated CD8 T cells, and type 1 T helper cells (Fig 3A). We also found that RA displayed a lower ssGSEA score of natural killer T cell in RA, which was consistent with the previous research [19] (Fig 3A).

Further, seven widely used algorithms were employed to calculate the infiltrating immune cell fraction in RA samples. In comparison to healthy samples, RA samples exhibited a significantly elevated proportion of monocyte, myeloid dendritic cell, activated myeloid dendritic cell, neutrophil and activated NK cell, in over two algorithms ($p < 0.05$, Wilcoxon rank-sum test, Fig 3B). In contrast, RA samples consistently exhibited a reduced proportion of naïve CD4+T cell across more than two algorithms ($p < 0.05$, Wilcoxon rank-sum test, Fig 3B). The Spearman correlation heatmap was plotted between immune cells and signature genes. As shown in Fig 3C, the expression level of *ICAM2* was negatively correlated with

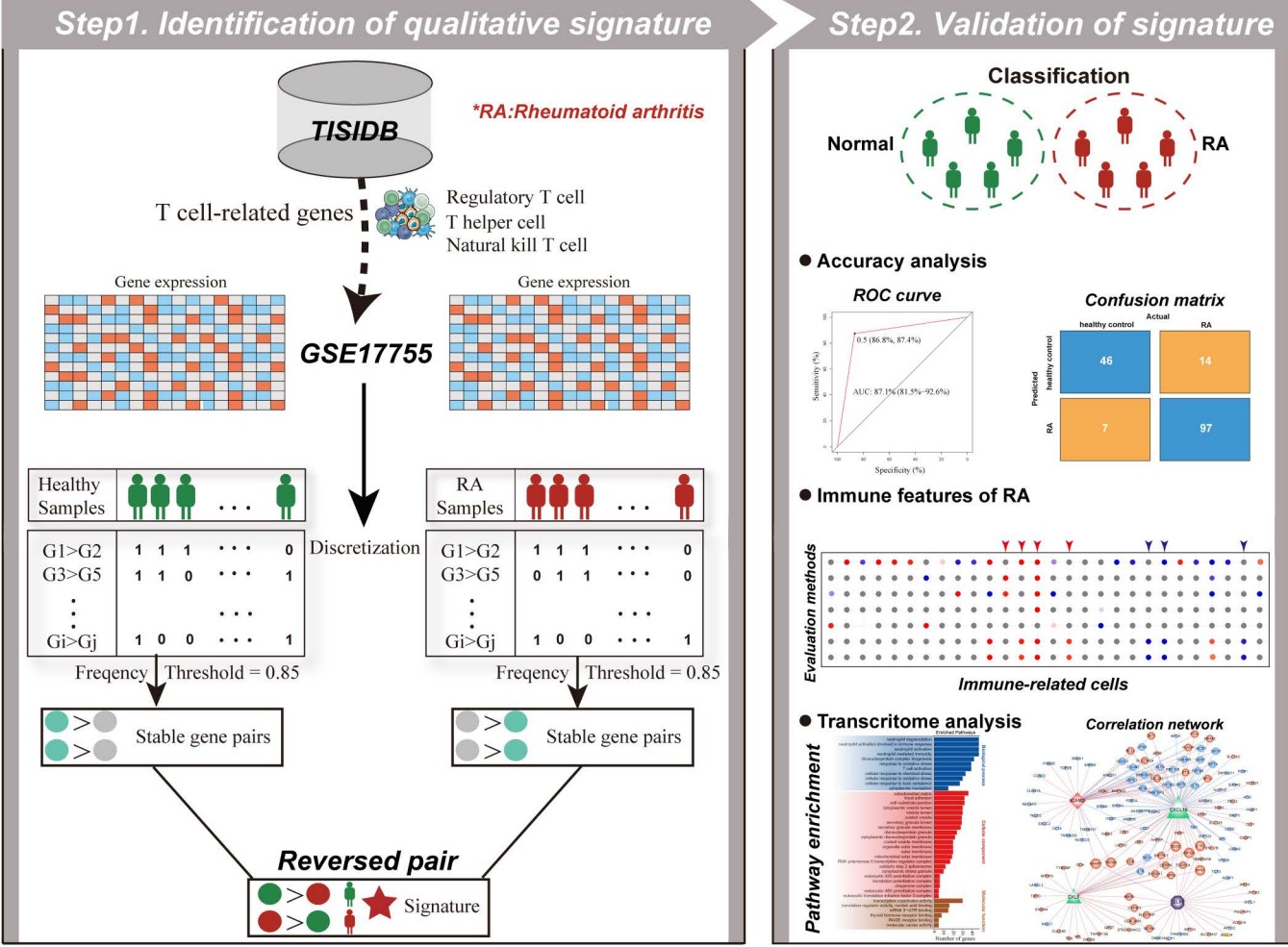

**Fig 1. Workflow for the identification and validation of qualitative signature in this study.**

the fraction of most immune cells, such as type 1 T helper cell, regulatory T cell and neutrophil. In contrast, there was a positive relationship between the expression of *OSTF1* and these immune cell infiltrates. Besides, IOS was found negative correlation with T cell infiltration (r = −0.36, spearman's rank correlation, S2 Fig). Furthermore, using Estimate method, we found the immune score in RA samples was significantly higher than in healthy samples (*p* < 0.001, Wilcoxon rank-sum test, Fig 3D). In summary, these results demonstrated that RA samples display an increased immune cell infiltration.

### RA-related genes were enriched with immune pathways

To investigate the dysregulated biological pathways related to RA, we performed pathway enrichment analysis on consistent genes dysregulated in RA across different datasets. Here, 4,030 and 4,107 differentially expressed genes (DEGs) were detected between RA and normal samples from the training and the integrated test datasets, respectively (FDR < 0.01, Wilcoxon rank-sum test). 800 of 1,454 DEGs were consistent between training and test datasets, which were defined as RA-related genes (Fig 4A). RA-related genes were significantly enriched in some pathways typically associated with immune microenvironment regulation processes (*p* < 0.05, Fig 4B), including "neutrophil degranulation", "neutrophil

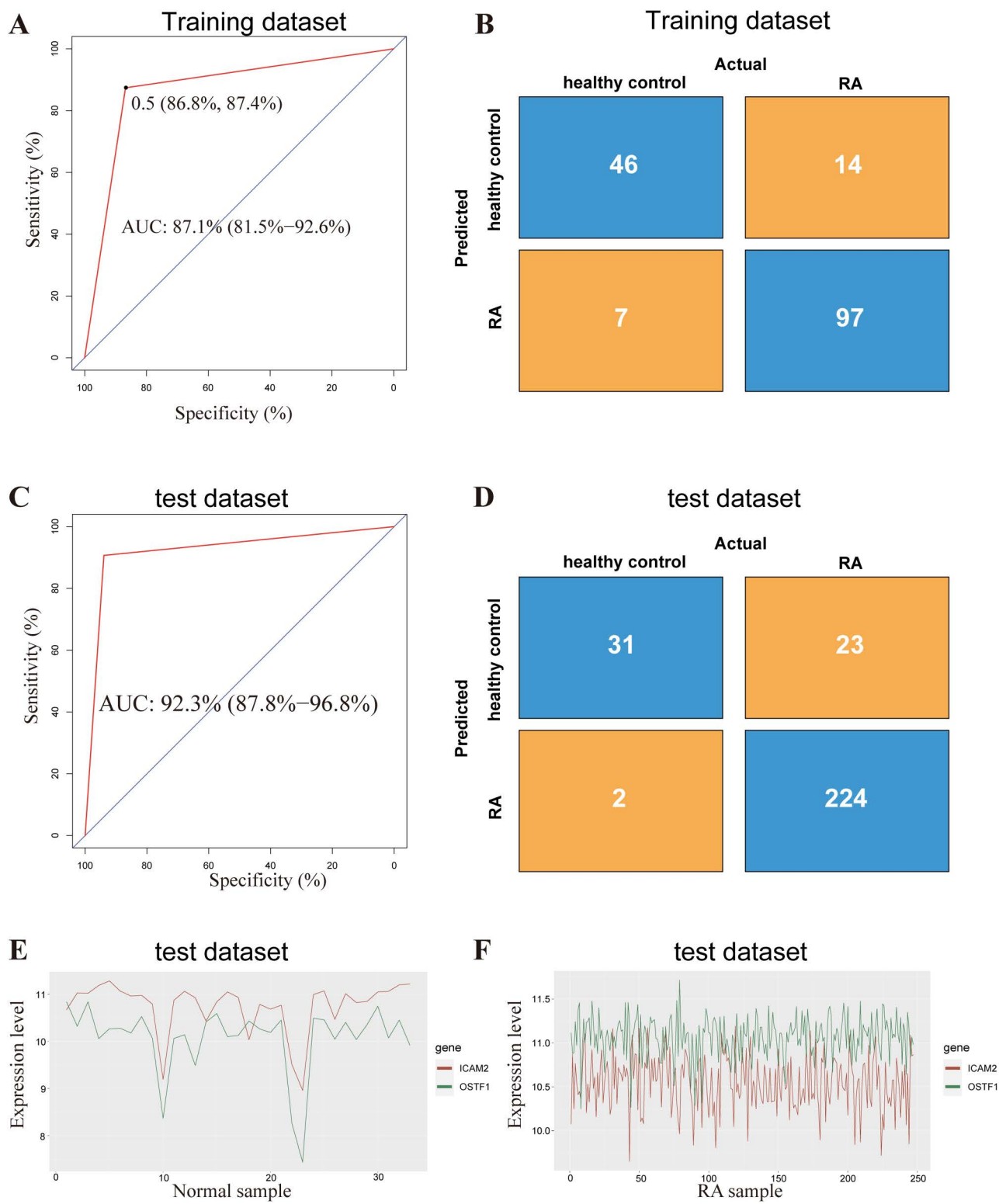

**Fig 2. Performance of IOS in the training and validation datasets.** (A-B) ROC curve and confusion matrix for the training dataset by IOS. (C-D) ROC curve and confusion matrix for the validating dataset by IOS. (E-F) The distribution of the quantitative expression levels of *ICAM2* and *OSTF1* in the validating dataset.

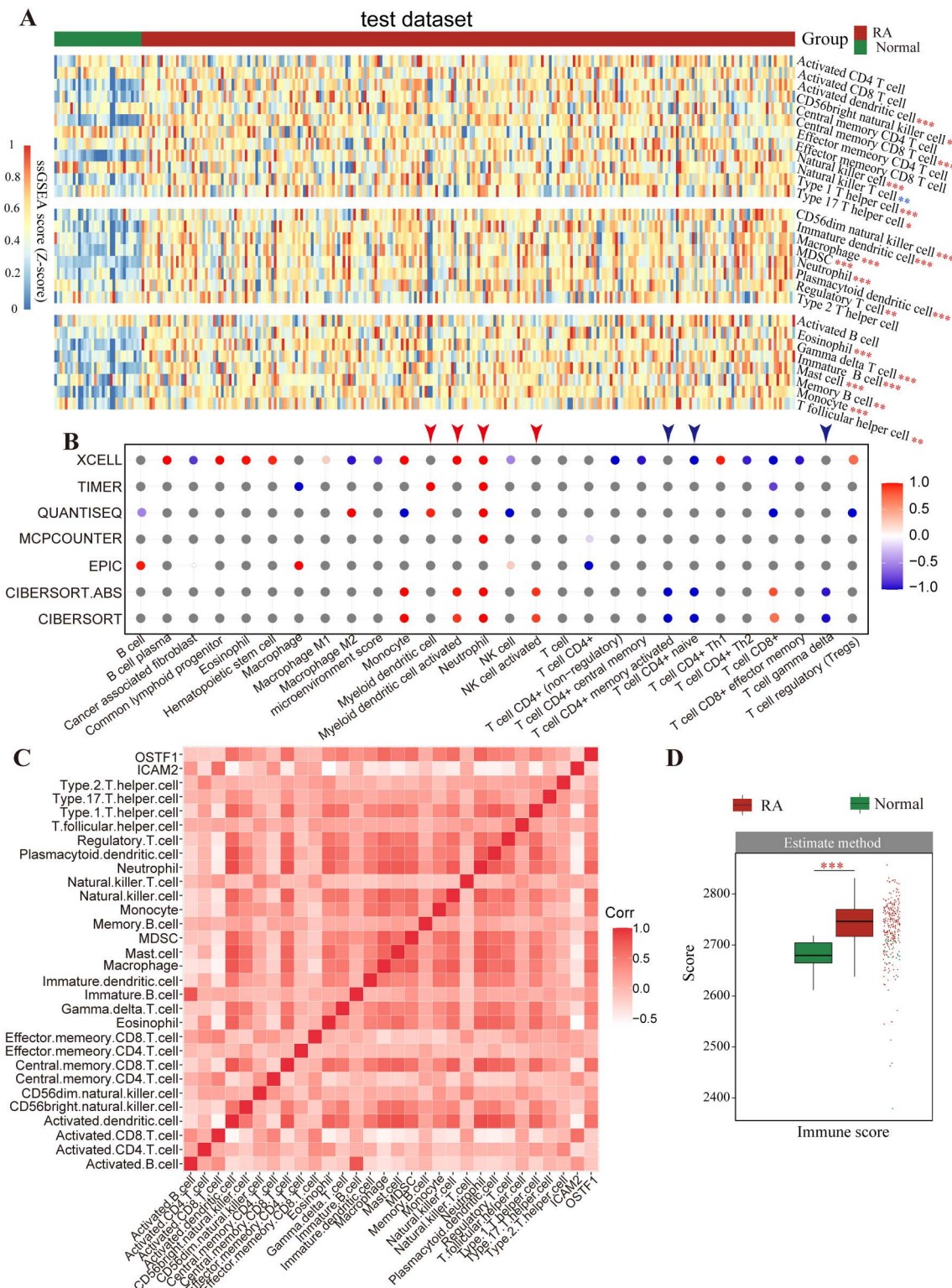

**Fig 3. Immune infiltration analysis between RA and normal samples.** (A) Identifying the relative infiltration of immune cell populations for 280 samples from the test dataset using ssGSEA method. The relative infiltration of each cell type is normalized with a z-score. (B) The fraction of immune cells in 280 samples is calculated based on seven algorithms in the test dataset. Immune cells with significantly higher (red dots) or lower (blue dots), or

without significant difference (gray dots) fractions in RA than normal samples were displayed. Wilcoxon rank-sum test, $p < 0.05$. The shade of the color represents the degree of significance. (C) The correlation heatmap among signature genes and fraction of immune cells was drawn. The shade of the color represents the degree and direction of correlation. The correlation coefficient was computed by Spearman rank correlation method. (D) Distribution of immune scores computed by Estimated method between RA and normal samples from the test dataset. $p$ values in the heatmap and box plot were calculated by one-sided Wilcoxon rank-sum test. *$P < 0.05$, **$P < 0.01$, ***$P < 0.001$.

activation involved in immune response", "neutrophil activation", "neutrophil mediated immunity" and "T cell activation" and so on. Similarly, using the Wilcoxon rank-sum test with FDR < 0.01 control, 683 IOS-related DEGs were identified between groups classified by IOS. And these DEGs participated T cell immune-related pathways such as "Human T-cell leukemia virus 1 infection" and "PD-L1 expression and PD-1 checkpoint pathway in cancer" ($p < 0.05$, S3 Fig) [32,33]. Additionally, when localizing inflammatory-related genes such as cytokine, inflammatory response, interferon response and chemokine, we found that some genes were also dysregulated in RA (Fig 4C-D). For example, compared with normal samples, *CKLF* had an increased expression level in RA, which was consistent with the previous reports [34]. As a gene related to inflammatory response, *JAK3* was a typical target treated with RA by JAK inhibitor, which was also overexpressed in RA [35]. These genes were associated with immune or inflammatory responses [36,37], thus possibly leading to the occurrence of RA.

### Network analysis of co-regulated genes by IOS

To investigate the dysregulated genes by RA, we performed Spearman correlation network analysis. RA-related genes and inflammation-related genes were used to construct the modulation network of RA. As shown in Fig 5A, the 43 RA-related genes regulated by *ICAM2* were significantly enriched in a biological pathway related to the occurrence and development of inflammatory arthritis, namely "Ferroptosis" [38] ($p < 0.05$, hypergeometric distribution model, Fig 5B). In addition, another enriched pathway, "Necroptosis", triggered innate immune responses, thus playing key roles in the inflammatory process (Fig 5B) [39]. Similarly, RA-related genes that interacted with *CXCL16* were found to be significantly enriched with inflammation- and/or immune-related pathways, such as "Ferroptosis", "Primary immunodeficiency" and "TNF signaling pathway" ($p < 0.05$, hypergeometric distribution model, Fig 5C). Using the hypergeometric distribution model and 33 RA-related genes linked with *CKLF* ($p < 0.05$), some inflammation- or immune-related pathways were also observed, such as "IL-17 signaling pathway" and "Fluid shear stress and atherosclerosis" (Fig 5D). Focusing on cytokine, we found *SLPI* was simultaneously connected to 24 RA-related genes, which affected some inflammation- or immune-related functions, such as "Fluid shear stress and atherosclerosis" and "Estrogen signaling pathway" ($p < 0.05$, hypergeometric distribution model, Fig 5E). As expected, these results suggest that the signature and inflammation-related genes are driving some RA-related transcriptome alterations. The relationship between IOS and these three genes was also evaluated. In addition to *SLPI*, a moderate positive correlation between IOS and the expression levels of other genes was achieved in training and test datasets ($r > 0.3$, spearman's rank correlation, S4 Fig).

### Discussion

In this study, we investigated the expression alterations of 233 T-cell related genes in the occurrence of RA and found that IOS could individually classify RA from normal peripheral blood samples. Samples were divided into RA group when the within-sample REO pattern was $E_{ICAM2} < E_{OSTF1}$; otherwise, the normal group. In comparison to typical quantitative signatures, qualitative signatures are extremely resistant to experimental technical changes and inadequate sample preparation [11,12]. And this signature can be used at the individual level without the need for a pre-collection of samples to establish a valid cutoff or data standardization [40]. For a total of 358 RA and 86 normal peripheral blood samples from GSE17755, GSE97475 and GSE97810, the overall sensitivity, specificity, and positive predictive value of the signature was 89.66%, 89.53% and 97.57%, indicating the robustness of IOS. It has been reported that

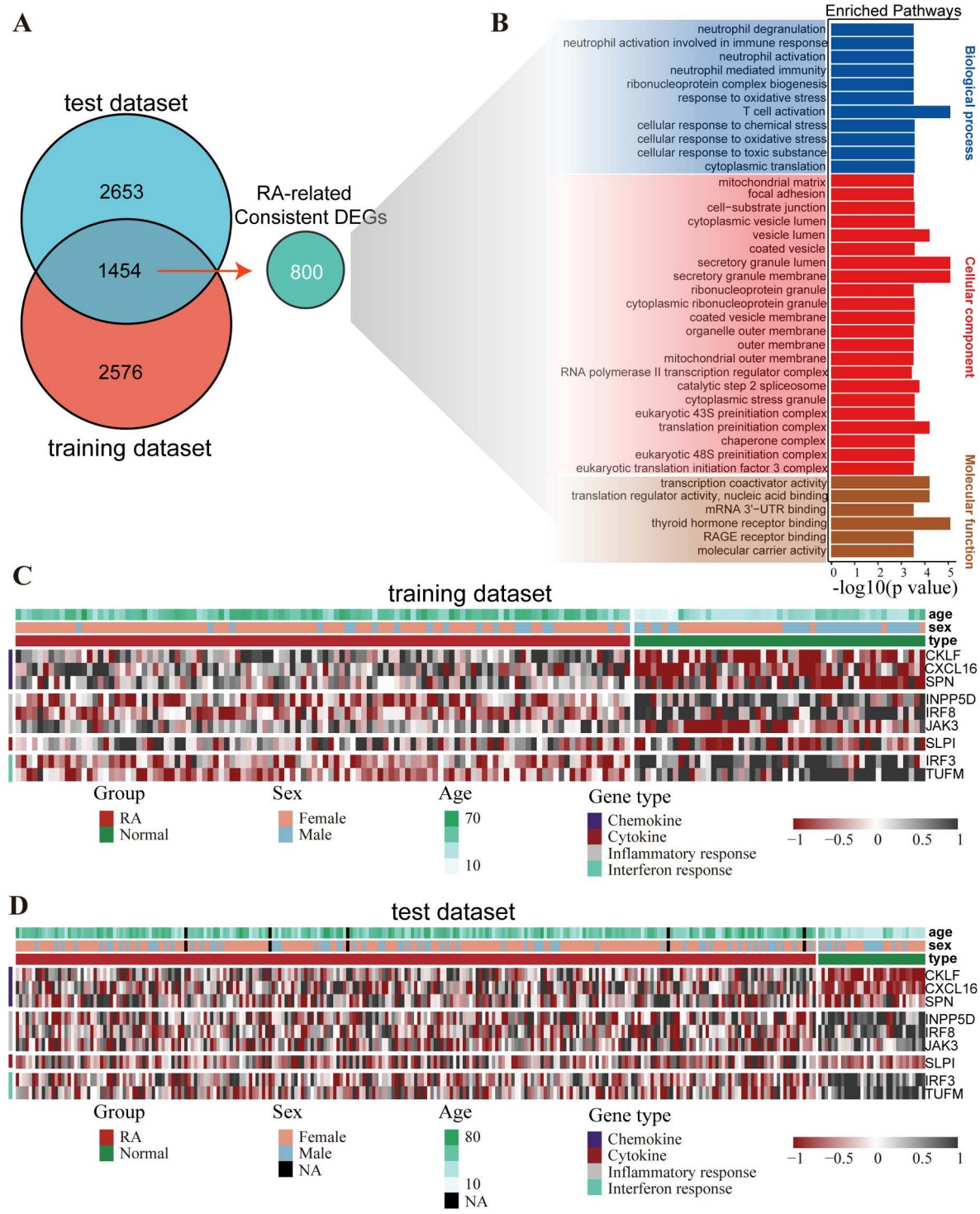

**Fig 4. Analysis of transcriptome differences between RA and normal samples.** (A) Venn diagram showing the overlap of RA-related DEGs between the training and test datasets. (B) GO pathways significantly enriched with 800 consistent DEGs were determined. *p* value was detected by hypergeometric distribution model and *p* < 0.05 was considered significant. Heatmap displaying the distribution for chemokine, cytokine, inflammatory response and interferon response genes with significant differences between the RA and normal samples in (C) training and (D) test datasets.

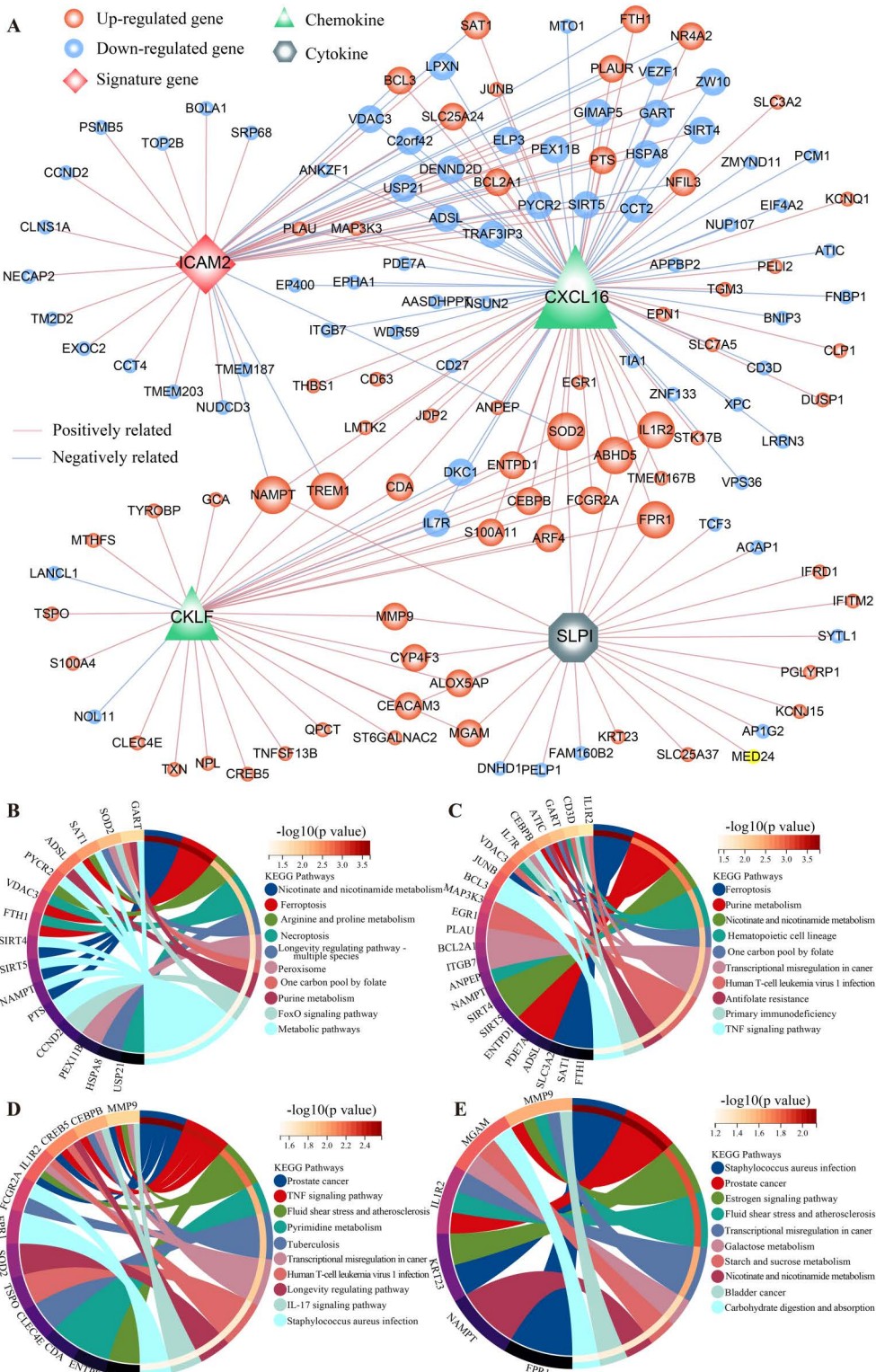

**Fig 5. The co-expression network of signature genes.** The nodes (circle) represent RA-related genes of co-expression with signature gene (rhombus), chemokine genes (trangle) and cytokine gene (hexagon); the blue and red lines represent negative and positive relationships, respectively. KEGG pathways significantly enriched with RA-related genes connected with *ICAM*, *CXCL16*, *CKLF* and *SLPI* are shown on (B), (C), (D) and (E), respectively.

the occurrence of RA was related to immune cell infiltration. The RA samples exhibited an increased abundance of five immune cell types, namely monocytes, myeloid dendritic cells, activated myeloid dendritic cells, neutrophils, and activated NK cells, while three immune cell types, specifically activated CD4 + memory T cells, naïve CD4 + T cells, and gamma delta T cells, showed a decreased presence. Besides, the immune score was significantly higher in RA when compared with the normal sample. Further functional enrichment analysis showed that RA-related dysregulated genes were enriched with some immune- and/or inflammation-related pathways. For instance, neutrophils are frequently detected in the synovial fluid in RA. Their presence in the synovial fluid is implicated in the induction of local inflammation, tissue damage, and erosion [41,42]. The "neutrophil activation" pathway could unleash proteolytic enzymes. These enzymes degrade extracellular matrix components in joints, harming cartilage and bone, and thus playing a significant role in the progression of RA [43,44]. "T cell activation" pathway can regulate the balance of different T-cell subsets such as helper T cells and regulatory T cells [45]. In patients with RA, there is usually an increase in type 17 T helper cells and impaired function of regulatory T cells [46]. Cytokines secreted by type 17 T helper cells, such as interleukin-17, can promote the inflammatory response. However, the abnormal function of regulatory T cells fails to effectively suppress the immune response, leading to immune imbalance and exacerbating the condition of RA [47,48]. Finally, we found that the network module driven by signature and inflammation-related genes might play a key role in RA-related transcriptional regulation.

Besides immune-related signature gene *ICAM2*, three key inflammation-related genes (*CXCL16*, *CKLF* and *SLPI*) were also major regulators in RA by modulating immune- and/or inflammation-related functions. CXC chemokine family genes are involved in inflammatory and immunological diseases, among which *CXCL16* influences the onset of inflammation via eliminating leukocyte chemotaxis, leukocyte adhesion, and endotoxin [49]. In addition, it has been revealed that chemokine gene *CKLF* is implicated in the pathogenic processes of RA and functionally relevant to inflammatory and immune response [50,51]. As a cytokine gene, *SLPI* promoted local tissue homeostasis by limiting innate immune cell damage and inhibiting the synthesis of pro-inflammatory cytokines, which leads to the recruitment of immune cells [52]. We also found that individual IOS status was positively correlated with these three key genes.

There are a few limitations in this study. Although our signature displayed a good performance in the testing dataset, it is necessary to collect more independent datasets with peripheral blood samples to validate the signature in the future. Besides, when focusing on the construction of the network, we were unable to verify the function of hub genes through biological experiments. However, enrichment analysis was used to investigate the pathways that these genes may affect.

In conclusion, the identified IOS might be used to classify RA from normal samples at the individual level, which would be useful in RA non-invasive diagnosis. As a result, IOS requires verification in a prospective clinical investigation to potentially minimize unnecessary costs associated with the early detection of RA.

## Supporting information

**S1 Fig. The distribution of the quantitative expression levels of *ICAM2* and *OSTF1* in the training dataset.** (PDF)

**S2 Fig. Relationship between IOS and T cell infiltration calculated by MCPCOUNTER.** (PDF)

**S3 Fig. Analysis of transcriptome differences between groups stratified by IOS.** Venn diagram showing the overlap of IOS-related DEGs between the training and test datasets. KEGG pathways significantly enriched with 683 consistent DEGs were determined. *p* value was detected by hypergeometric distribution model and $p < 0.05$ was considered significant. (PDF)

**S4 Fig. Relationship between IOS and the expression of *CXCL16*, *CKLF*, and *SLPI* in (A) training dataset and (B) test dataset, respectively.**
(PDF)

**S1 Table. Inflammatory-related gene lists used in this study.**
(PDF)

## Author contributions

**Conceptualization:** Hang Su, Yawei Li.

**Data curation:** Xingyi Li.

**Funding acquisition:** Yawei Li.

**Investigation:** Xingyi Li.

**Methodology:** Hang Su, Xingyi Li.

**Project administration:** Yawei Li.

**Software:** Hang Su, Xingyi Li.

**Supervision:** Yawei Li.

**Validation:** Hang Su, Xingyi Li.

**Visualization:** Xingyi Li.

**Writing – original draft:** Hang Su, Yawei Li.

**Writing – review & editing:** Hang Su, Yawei Li.

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
