## [Decision Letter · Decision Letter 0]

Dear Dr. Li,

Thank you for submitting your manuscript to PLOS ONE. After careful consideration, we feel that it has merit but does not fully meet PLOS ONE’s publication criteria as it currently stands. Therefore, we invite you to submit a revised version of the manuscript that addresses the points raised during the review process.

We look forward to receiving your revised manuscript.

Kind regards,

Austin W.T. Chiang

Academic Editor

PLOS ONE

**Journal Requirements:**

https://wjso.biomedcentral.com/articles/10.1186/s12957-024-03314-8

In your revision ensure you cite all your sources (including your own works), and quote or rephrase any duplicated text outside the methods section. Further consideration is dependent on these concerns being addressed.

4. Please note that your Data Availability Statement is currently missing the direct link to access each database. If your manuscript is accepted for publication, you will be asked to provide these details on a very short timeline. We therefore suggest that you provide this information now, though we will not hold up the peer review process if you are unable.

Reviewers' comments:

Reviewer's Responses to Questions

**Comments to the Author**

1. Is the manuscript technically sound, and do the data support the conclusions?

Reviewer #1: Yes

Reviewer #2: Partly

Reviewer #3: Yes

2. Has the statistical analysis been performed appropriately and rigorously?

Reviewer #1: Yes

Reviewer #2: Yes

Reviewer #3: Yes

3. Have the authors made all data underlying the findings in their manuscript fully available?

Reviewer #1: Yes

Reviewer #2: Yes

Reviewer #3: Yes

4. Is the manuscript presented in an intelligible fashion and written in standard English?

Reviewer #1: Yes

Reviewer #2: Yes

Reviewer #3: No

**Reviewer #1:**  Introduction

1. Why do the serum biomarkers currently used to detect RA in clinical practice lack sufficient accuracy? Can these biomarkers be used to explain the heterogeneous features of RA patients? If so, how can their accuracy be improved?

2. How can signatures developed based on within-sample relative expression orderings (REOs) of gene pairs be used to explain the heterogeneous features of RA patients?

3. Regarding the role of T cells, can the author explain more clearly why T cells are necessary for the onset and progression of RA?

Materials and Methods

1. Concerning the mapping of probe set IDs, has the author considered about calculating a gene’s expression value by using its maximum value among the corresponding probe sets? If so, what were the effects?

2. What are the advantages of the REO method compared to traditional correlation-based methods?

3. Can the REO method be applied to certain aspects of immunological feature analysis and network analysis?

Results

1. Why does only one gene pair, ICAM2-OSTF1, overlap between TRGPs and the reversal gene pairs? Could other results be obtained if the methods were optimized?

2. The sentence "the REO patterns of IOS are stably" should be corrected to "the REO patterns of IOS are stable."

3. Could the author provide more details on the correlation between ICAM2-OSTF1 and T-cell infiltration?

4. Could the author provide more details on the correlation between ICAM2-OSTF1 and T-cell immune pathways?

Discussion

1. How can the REO method outperform other approaches in explaining the heterogeneous features of RA patients?

2. Why are the three key inflammation-related genes (CXCL16, CKLF, and SLPI) discussed? What is their relationship with ICAM2-OSTF1?

3. Could the author describe the biological significance of the ICAM2-OSTF1-related network, and how it can be validated experimentally?

**Reviewer #2:**  1. In Materials and Methods section, in terms of the validation dataset, the normal sample and RA sample are from two dataset. There will be potential bias on these dataset design and batch effect. How do you address these problems?

2. While the validation datasets are independent, they originate from similar sources (peripheral blood mononuclear cells). If possible, including other type of dataset could further validate your model’s robustness.

Minor issues

1. Line 150, “97 of 111 RA samples”, is total should be 112?

2. For the figures, it will be better to save them in higher resolution files.

In conclusion, the paper introduces a novel diagnostic framework with potential clinical impact. Addressing the comments above would enhance its translational application in RA diagnosis.

**Reviewer #3: ** Summary

This research introduces a diagnostic signature for rheumatoid arthritis (RA) based on a rank-based qualitative method utilizing T-cell-related gene expression profiles. The identified ICAM2-OSTF1 signature (IOS) demonstrates high sensitivity, specificity, and accuracy in distinguishing RA from healthy samples across multiple datasets. The study also investigates immune infiltration and transcriptional differences in RA, linking these finding to immune microenvironmental changes. The results provide a robust and non-invasive diagnostic tool while suggesting potential molecular targets involved in RA pathogenesis. Future studies should validate the ICAM2-OSTF1 signature in larger and more diverse datasets. Additionally, the functional role of ICAM2 and OSTF1 should be experimentally investigated.

Comments

The manuscript is logically well-structured and presents its finding effectively. However, it contains many typos and grammatical errors, some of which have been identified. To improve the overall quality and readability, the language should be carefully revised and polished, preferably by a native English speaker.

1. In line 58, the phrase “immunity balanced” is grammatically incorrect. It should be revised to “immune balance”. Please consider rephrasing as “whose dysfunction contributes…” instead of “the dysfunction of which contributes…” for improved clarity.

2. In line 78, “which was including” is grammatically incorrect. Please replace it with “Which includes” for better grammar and readability.

3. The quality of the figures needs improvement. Some of them are not readable. Please ensure they are clear and legible.

4. In the sentence spanning lines 121 to 123, there are multiple grammar error, please rephrase it for clarity.

5. Delete first “and” in line 122.

6. Figure 2F is not mentioned in the manuscript. Please ensure that all figures are properly described and mentioned with in the text.

7. The X-axis label in Figure 2F is “Tumor sample”, but I did not see any tumor samples in the manuscript. Should this be corrected to “RA samples”?

8. In line 172, replace “naiver” change to “naïve”.

9. In Figure 3B, monocytes and activated NK cells appear to increase in RA samples, but the author did not mention them in results section. Similarly, decreased fractions like activated CD4 memory T cells and gamma delta T cells were not discussed. These findings should be included for completeness.

10. Figure 3, 4 and 5 are not readable in its current form. Enhance its quality for clarity and ensure all labels and data are visible.

11. The term "800 consistent DEGs" needs definition. Do these include only upregulated genes, or do they encompass both upregulated and downregulated genes?

12. In Figure 4B, please consider using the adjusted p-value on the X-axis instead of the number of genes to enhance interpretability.

13. In Figure 4, several immune-related pathways (e.g., neutrophil activation and T-cell activation) are mentioned, but the manuscript lacks further discussion on how these pathways influence the onset and progression of RA. Please either conduct a more detailed analysis of these pathways or reference existing literature discussing their roles in RA.

14. In lines 190-191, clarity how the authors localized inflammatory-related genes. Was this derived from KEGG analysis? I think your input genes for KEGG analysis are dysregulated gene already. Please clarify.

15. From lines 193 to 196, the author mentioned two RA related Genes CKLF and JAK3, then the author state that these genes are associated with dysfunction of the immune microenvironment in RA. However, there is no evidence presented on how these genes affect the immune microenvironment. Please provide references to relevant studies and avoid over-concluding.

16. In line 218, the mention of ion channel genes is unrelated and confusing, as these genes are not discussed elsewhere in the manuscript. Clarify or remove this statement.

**Do you want your identity to be public for this peer review?** For information about this choice, including consent withdrawal, please see our Privacy Policy

Reviewer #1: No

Reviewer #2: No

Reviewer #3: **Yes: ** Qingkang Lyu

---

## [Author Response · Author response to Decision Letter 1]

4 Mar 2025

Journal Requirements:

Reply 1: Thanks. Done as the requested.

https://wjso.biomedcentral.com/articles/10.1186/s12957-024-03314-8

In your revision ensure you cite all your sources (including your own works), and quote or rephrase any duplicated text outside the methods section. Further consideration is dependent on these concerns being addressed.

Reply 2: Thanks. The description that may involve overlap has been modified. Correction are as follows.

In the revised Materials and methods in lines 83-88: “The same preprocessing was performed for these datasets, the gene expression profiles (series.txt) were directly used after the process of probe re-annotation without any normalization. The probe set IDs were mapped to Entrez gene IDs with the corresponding platform annotation file for all gene expression profile data. If multiple probe sets were mapped to the same gene, the expression value of the gene was calculated by its average value. Besides, if a probe mapped multiple Entrez gene IDs or did not map any Entrez gene ID, the probe was deleted.” � “The identical preprocessing was executed for these datasets, with the gene expression profiles (series.txt) utilized directly following the probe re-annotation process, devoid of any normalization. The probe set IDs were correlated with Entrez gene IDs using the relevant platform annotation file for all gene expression profile data. If numerous probe sets corresponded to the same gene, the expression value was determined by calculating its average value. Furthermore, probes that mapped multiple Entrez gene IDs or failed to map any Entrez gene ID were eliminated.”;

In the revised Results in lines 157-158: “With the majority voting rule, 97 of 112 RA samples and 46 of 53 healthy samples in the training dataset were correctly classified” �“97 out of 112 RA samples and 46 out of 53 healthy samples were accurately categorized.”;

In the revised Results in lines 175-178: “It is known that the occurrence and development of RA are related to immune microenvironments. To determine the relation between RA and immune microenvironments, we composed a heatmap to visualize the abundance of 28 infiltrating immune cell populations of 280 samples from the test datasets using the ssGSEA method.” � “The occurrence and development of RA are associated with immunological microenvironments. Utilizing the ssGSEA approach, the abundance of 28 infiltrating immune cell types was illustrated using 280 samples from the testing datasets to investigate the correlation between RA and microenvironments.”;

In the revised Results in lines 178-179: “Results showed that RA had a higher abundance of immune cell infiltration (e.g., activated dendritic cell, activated CD8 T cell, and Type 1 T helper cell, Figure 3A).” � “The results indicated that RA exhibited a greater abundance of immune cell infiltration, including activated dendritic cells, activated CD8 T cells, and type 1 T helper cells (Fig 3A).”;

In the revised Results in lines 182-183: “Here, we integrated seven popular algorithms to estimate the infiltrating immune cell fraction in RA samples.” � “Further, seven widely used algorithms were employed to calculate the infiltrating immune cell fraction in RA samples.”;

In the revised Results in line 183: “Compared with healthy samples” � “In comparison to healthy samples”; “showed” � “exhibited”; “higher fraction” �“elevated proportion”;

In line 184: “On the contrary”�“In contrast”;

In lines 185-186: “showed a lower fraction of” � “exhibited a reduced proportion of”; “, in over two algorithms” � “across more than two algorithms”;

In the revised Discussion in lines 310-312: “IOS should be verified in a prospective clinical investigation, which might reduce unnecessary expenses for the early detection of RA.” � “IOS requires verification in a prospective clinical investigation to potentially minimize unnecessary costs associated with the early detection of RA.”

Reply 3: Thanks. Done as the requested. We provided the correct funding information in the Funding section in lines 333-335: “This research was supported by a grant from the Basic Research Program of Guizhou Science and Technology Department (ZK[2024]156) and the High-level Talents Startup Fund of Guizhou Medical University (J[2022]042).”

4. Please note that your Data Availability Statement is currently missing the direct link to access each database. If your manuscript is accepted for publication, you will be asked to provide these details on a very short timeline. We therefore suggest that you provide this information now, though we will not hold up the peer review process if you are unable.

Reply 2: Thanks. We have added the direct link of data used in the present study. In the revised Data availability: “The datasets analyzed during the present study are available from the Gene Expression Omnibus (https://www.ncbi.nlm.nih.gov/geo/query/acc.cgi?acc=GSE17755;
https://www.ncbi.nlm.nih.gov/geo/query/acc.cgi?acc=GSE97475;https://www.ncbi.nlm.nih.gov/geo/query/acc.cgi?acc=GSE97810)”

Reply 5: Thanks. Done as the requested.

Review Comments to the Author

Reviewer #1:

Introduction

1. Why do the serum biomarkers currently used to detect RA in clinical practice lack sufficient accuracy? Can these biomarkers be used to explain the heterogeneous features of RA patients? If so, how can their accuracy be improved?

Reply 1: Thanks for your comments. Rheumatoid factor (RF) and Anti-cyclic citrullinated peptide antibody (ACPA) are currently used for RA diagnosis in the clinic. However, they also can be found in other connective tissue diseases, some chronic infectious diseases, or systemic lupus erythematosus, leads to a poor accuracy [1, 2]. Besides, since the diagnosis of RA mainly relies on the expression measurement of serum biomarkers, batch effects may also affect the test results [3]. They do explain, to some extent, the heterogeneity of RA. For example, patients with higher titers of RF and anti-CCP antibodies tend to have more severe joint destruction and may progress more quickly [4, 5]. This suggests that there is heterogeneity in the severity of the disease among patients with different antibody levels. Therefore, a potential method to improve their accuracy is to combine with indicators related to the degree of inflammatory activity of the disease such as C-reactive protein and erythrocyte sedimentation rate for RA diagnosis.

2. How can signatures developed based on within-sample relative expression orderings (REOs) of gene pairs be used to explain the heterogeneous features of RA patients?

Reply 2: Thanks for your comments. The gene pair signature is not used to explain the heterogeneous features of RA patients but to overcome the batch effects across heterogeneous patient cohorts, which is a major obstacle to RA early detection.

We have re-described in Introduction in lines 50-53: “In contrast, signatures developed based on within-sample relative expression orderings (REOs) of gene pairs are resistant to batch effects across heterogeneous patient cohorts or normalization effects by different technical sources and can be applied to individual samples [6-8].”

3. Regarding the role of T cells, can the author explain more clearly why T cells are necessary for the onset and progression of RA?

Reply 3: Thanks. The immunomodulatory imbalance induced by T cells also promotes the occurrence and progression of RA. For example, in healthy humans, Th1 and Th2 cells are in balance to maintain immunological homeostasis. However, Th1 cells are predominant in RA patients. The increased secretion of cytokines such as interferon-γ by Th1 cells can promote the activation of immune cells such as macrophages, enhance cellular immune response, lead to aggravated inflammatory response, and promote the occurrence and development of RA [9]. Besides, the balance between Th17 and Treg cells is also essential for maintaining immune tolerance and immune homeostasis. In RA, Th17 cell function is enhanced and Treg cell function is relatively insufficient. This imbalance prevents the body from properly regulating the immune response against its own joint tissues, which results in RA [10].

Materials and Methods

1. Concerning the mapping of probe set IDs, has the author considered about calculating a gene’s expression value by using its maximum value among the corresponding probe sets? If so, what were the effects?

Reply 1: Thanks, in probe re-annotation, individual extreme values may appear due to the experimental errors, if the maximum value is used, these outliers may have a greater impact on the results, leading to an incorrect estimate of the gene expression level. The average value can reduce the interference of outliers to a certain extent by averaging all probes, and more stably reflect the true expression of genes. Therefore, we used the average value rather than the maximum value.

2. What are the advantages of the REO method compared to traditional correlation-based methods?

Reply 2: Thanks, REO method, unlike traditional correlation-based methods, has been reported that: (i) is resistant to experimental and technical variations; (ii) is invariant to monotonic data normalization; (iii) can be applied to individual samples without a complex scoring process [3, 11].

3. Can the REO method be applied to certain aspects of immunological feature analysis and network analysis?

Reply 3: Thanks. At present, our laboratory uses the REO method, considering its advantages, mainly in three aspects: (i) the discovery of disease biomarkers [7, 8]; (ii) the identification of individual pathways [12]; (iii) the identification of individualized differentially expressed genes [13]. (ii) and (iii) may be involved in immunological feature analysis and network analysis.

Results

1. Why does only one gene pair, ICAM2-OSTF1, overlap between TRGPs and the reversal gene pairs? Could other results be obtained if the methods were optimized?

Reply 1: This result mainly depends on the data we used in the present study, especially since only T cell-related gene expression profiles were involved. In this context, even if alternative gene pairs are identified through method optimization, their diagnostic performance is inferior to that of the ICAM2 - OSTF1.

2. The sentence "the REO patterns of IOS are stably" should be corrected to "the REO patterns of IOS are stable."

Reply 2: Thanks. Done as suggested.

3. Could the author provide more details on the correlation between ICAM2-OSTF1 and T-cell infiltration?

Reply 3: Thanks. Done as suggested. We have added the related result in the revised Result in lines 190-191: “Besides, ICAM2-OSTF1 was found negative correlation with T cell infiltration (r=-0.36, spearman’s rank correlation, S2 Fig).”

S2 Fig. Relationship between ISO and T cell infiltration calculated by MCPCOUNTER.

4. Could the author provide more details on the correlation between ICAM2-OSTF1 and T-cell immune pathways?

Reply 4: Thanks. We could not directly evaluate the correlation because the individual pathway scores cannot be quantitative. However, using the wilcoxon rank-sum test with FDR < 0.01 control, 683 IOS-related DEGs were identified between groups classified by IOS. And these DEGs participated T cell immune-related pathways such as “Human T-cell leukemia virus 1 infection” and “PD-L1 expression and PD-1 checkpoint pathway in cancer” [14, 15]. The related descriptions are in lines 218-221: “Similarly, using the wilcoxon rank-sum test with FDR < 0.01 control, 683 IOS-related DEGs were identified between groups classified by IOS. And these DEGs participated T cell immune-related pathways such as “Human T-cell leukemia virus 1 infection” and “PD-L1 expression and PD-1 checkpoint pathway in cancer” (p<0.05, S3 Fig)”

S3 Fig.Venn diagram showing the overlap of IOS-related DEGs between the training and test datasets. KEGG pathways significantly enriched with 683 consistent DEGs were determined. p value was detected by hypergeometric distribution model and p < 0.05 was considered significant.

Discussion

1. How can the REO method outperform other approaches in explaining the heterogeneous features of RA patients?

Reply 1: Thanks. Please see Reply 2 in the Introduction.

2. Why are the three key inflammation-related genes (CXCL16, CKLF, and SLPI) discussed?

What is their relationship with ICAM2-OSTF1?

Reply 2: Thanks. CXCL16, CKLF, and SLPI are with significantly differentially expressed inflammation-related genes between RA and normal sample with large degrees in the network (Fig 5), which are more possibility to have potential regulatory effects on RA occurrence or development. According to the reviewer’s suggestion, we evaluated their relationship. We have added relation descriptions in the revised Results in lines 255-257: “The relationship between IOS and these three genes was also evaluated. In addition to SLPI, a moderate positive correlation between IOS and the expression levels of other genes was achieved in training and test datasets (r>0.3, spearman’s rank correlation, S4 Fig).” and in the revised Discussion in lines 302-303: “We also found that individual IOS status was positively correlated with these three key genes.”

S4 Fig. Relationship between ISO and the expression of CXCL16, CKLF, and SLPI in (A) training dataset and (B) test dataset, respectively.

3. Could the author describe the biological significance of the ICAM2-OSTF1-related network, and how it can be validated experimentally?

Reply 3: Thanks. In the present study, this network was constructed based on the correlation matrix of expression, thus we cannot obtain the direct regulatory relationship between these differentially expressed genes. However, as described in Reply 2, we found four genes (CXCL16, CKLF, SLPI and ICAM2) achieved a large degree in the network, which were considered as potential regulators of RA occurrence and development. Currently, the conditions for functional verification of hub genes are not yet available, which requires animal models, cell models, knockout and other experimental technologies. We have discussed the related description in the limitation of the study in lines 306-308: “Besides, when focusing on the construction of the network, we were unable to verify the function of hub genes through biological experiments. However, enrichment analysis was used to investigate the pathways that these genes may affect.”

Reviewer #2:

1. In Materials and Methods section, in terms of the validation dataset, the normal sample and RA sample are from two dataset. There will be potential bias on these dataset design and batch effect. How do you address these problems?

Reply 1: Thanks for your comments. The REO method we used in this study is resistant to batch effects or normalization effects by different technical sources and can be applied to individual samples. The related description is in the Introduction in lines 51-54.

2. While the validation datasets are independent, they originate from similar sources (peripheral blood mononuclear cells). If possible, including other type of dataset could further validate your mo

---

## [Decision Letter · Decision Letter 1]

Dear Dr. Li,

Thank you for submitting your manuscript to PLOS ONE. After careful consideration, we feel that it has merit but does not fully meet PLOS ONE’s publication criteria as it currently stands. Therefore, we invite you to submit a revised version of the manuscript that addresses the points raised during the review process.

We look forward to receiving your revised manuscript.

Kind regards,

Wan-Tien Chiang

Academic Editor

PLOS ONE

Journal Requirements:

Reviewers' comments:

Reviewer's Responses to Questions

**Comments to the Author**

Reviewer #1: All comments have been addressed

Reviewer #2: All comments have been addressed

Reviewer #3: (No Response)

2. Is the manuscript technically sound, and do the data support the conclusions?

Reviewer #1: Yes

Reviewer #2: Yes

Reviewer #3: Yes

3. Has the statistical analysis been performed appropriately and rigorously?

Reviewer #1: Yes

Reviewer #2: Yes

Reviewer #3: Yes

4. Have the authors made all data underlying the findings in their manuscript fully available?

Reviewer #1: Yes

Reviewer #2: Yes

Reviewer #3: Yes

5. Is the manuscript presented in an intelligible fashion and written in standard English?

Reviewer #1: Yes

Reviewer #2: Yes

Reviewer #3: Yes

Reviewer #1: The corrections fully support a value of the method----within-sample relative expression orderings (REOs) of gene pairs for diagnosis of rheumatoid arthritis. All parts in this manuscript which should be corretced have been corrected.

Reviewer #2: (No Response)

Reviewer #3: Thank you for your prompt response and revision. Most of my concerns have been addressed, and the manuscript has been significantly improved. However, I still have two remaining concerns:

1. In line 58, the phrase "immunity balanced" is grammatically incorrect and has not been corrected. Please replace it with "immune balance."

2. In response to comment 12, the authors have provided a high-resolution figure for Figure 4. However, I did not notice any changes to Figures 3 and 5, as mentioned in comment 10.

**Do you want your identity to be public for this peer review?** For information about this choice, including consent withdrawal, please see our Privacy Policy

Reviewer #1: No

Reviewer #2: No

Reviewer #3: No

---

## [Author Response · Author response to Decision Letter 2]

11 May 2025

Review Comments to the Author

Reviewer #1: The corrections fully support a value of the method----within-sample relative expression orderings (REOs) of gene pairs for diagnosis of rheumatoid arthritis. All parts in this manuscript which should be corretced have been corrected.

Reply: We thank the reviewer for the agreement with the main findings of our work.

Reviewer #2: (No Response)

Reviewer #3: Thank you for your prompt response and revision. Most of my concerns have been addressed, and the manuscript has been significantly improved. However, I still have two remaining concerns:

1. In line 58, the phrase "immunity balanced" is grammatically incorrect and has not been corrected. Please replace it with "immune balance."

Reply: We would like to express our sincere appreciation for your help to our manuscript. "immunity balanced" have been replaced with "immune balance."

2. In response to comment 12, the authors have provided a high-resolution figure for Figure 4. However, I did not notice any changes to Figures 3 and 5, as mentioned in comment 10.

Reply: To make the Figures more legible, we have increased the font size of 3B-C and 5B-E. The resolution of figures 3 and 5 has also changed from 400 ppi to 600 ppi. It is possible that the PDF document generation system causes the figures to appear unclear, but this is clearly stated in the doc file. I have listed the revised figures 3 and 5 below.

---

## [Editor Report · Decision Letter 2]

Individualized diagnosis of rheumatoid arthritis: a rank-based qualitative T cell-related signature

PONE-D-24-53992R2

Dear Dr. Li,

We’re pleased to inform you that your manuscript has been judged scientifically suitable for publication and will be formally accepted for publication once it meets all outstanding technical requirements.

Kind regards,

Austin W.T. Chiang

Academic Editor

PLOS ONE

---

## [Editor Report · Acceptance letter]

PONE-D-24-53992R2

PLOS ONE

Dear Dr. Li,

I'm pleased to inform you that your manuscript has been deemed suitable for publication in PLOS ONE. Congratulations! Your manuscript is now being handed over to our production team.

Kind regards,

on behalf of

Dr. Wan-Tien Chiang

Academic Editor

PLOS ONE